# Serological Surveillance of COVID-19 Hospitalized Patients in Réunion Island (France) Revealed that Specific Immunoglobulin G Are Rapidly Vanishing in Severe Cases

**DOI:** 10.3390/jcm9123847

**Published:** 2020-11-27

**Authors:** Anthony Dobi, Anne-Laure Sandenon Seteyen, Mahary Lalarizo Rakoto, Grégorie Lebeau, Damien Vagner, Étienne Frumence, Claude Giry, Axelle Septembre-Malaterre, Loïc Raffray, Philippe Gasque

**Affiliations:** 1Unité de Recherche en Pharmaco-Immunologie (UR-EPI), Université et CHU de La Réunion, 97400 Saint-Denis, France; anne.sandenon@gmail.com (A.-L.S.S.); mahary11@gmail.com (M.L.R.); greg.lebeau@live.fr (G.L.); etienne.frum@gmail.com (É.F.); claude.giry@chu-reunion.fr (C.G.); axelle.malaterre-septembre@univ-reunion.fr (A.S.-M.); loic.raffray@chu-reunion.fr (L.R.); philippe.gasque@univ-reunion.fr (P.G.); 2Faculté de Médecine, Campus universitaire Ambohitsaina, BP375, 101 Antananarivo, Madagascar; 3UMR PIMIT ‘Processus Infectieux en Milieu Insulaire Tropical’ CNRS 9192, INSERM1187, IRD 249, Université de La Réunion, 97400 Saint-Denis, France; damien.vag@gmail.com; 4CNR Associé Arbovirus, Laboratoire de Microbiologie, Pôle de Biologie, CHU de La Réunion, 97440 Saint-Denis, France; 5Service de Médecine Interne, CHU de La Réunion, 97400 Saint-Denis, France; 6Laboratoire d’Immunologie Clinique et Expérimentale de la ZOI (LICE-OI), Pôle de Biologie, CHU de La Réunion, 97400 Saint-Denis, France

**Keywords:** COVID-19, disease severity, intensive care unit, immunoglobulin G, serological analyses, enzyme-linked immunosorbent assay, Western blot

## Abstract

Humoral immunity is critically important to control COVID-19. Long-term antibody responses remain to be fully characterized in hospitalized patients who have a high risk of death. We compared specific Immunoglobulin responses against severe acute respiratory syndrome coronavirus 2 (SARS-CoV-2) between two groups, intensive care unit (ICU) and non-ICU hospitalized patients over several weeks. Plasma specific IgG, IgM, and IgA levels were assessed using a commercial ELISA and compared to an in-house cell-based ELISA. Among the patients analyzed (mean (SD) of age, 64.4 (15.9) years, 19.2% female), 12 (46.2%) were hospitalized in ICU. IgG levels increased in non-ICU cases from the second to the eighth week after symptom onset. By contrast, IgG response was blunted in ICU patients over the same period. ICU patients with hematological malignancies had very weak or even undetectable IgG levels. While both groups had comparable levels of specific IgM antibodies, we found much lower levels of specific IgA in ICU versus non-ICU patients. In conclusion, COVID-19 ICU patients may be at risk of reinfection as their specific IgG response is declining in a matter of weeks. Antibody neutralizing assays and studies on specific cellular immunity will have to be performed.

## 1. Introduction

According to the World Health Organization (WHO), the coronavirus disease 2019 (COVID-19) pandemic currently affects more than 48 million individuals (confirmed cases) worldwide. Clinical features of COVID-19 range from asymptomatic or non-severe disease to critically ill-patients with life-threatening conditions. Non-severe cases display symptoms such as fever, dry cough, and fatigue, but have a good prognosis. By contrast, up to 10–15% of hospitalized patients progress to a severe form of the disease and may develop pneumonia, acute respiratory distress syndrome, and multiple organ failure, requiring long stays in intensive care units (ICU) [1,2,3].

Long-term humoral response mediated by Immunoglobulin (Ig) is critically important to control a viral infection. Previous investigations described significantly different IgG levels in severe/critical cases as compared to non-severe/mild cases during the early phases of COVID-19 [4,5]. This difference may be due to the fact that infected individuals may not have comparable antigen loads and uniform antibody responses over the time of the disease, as suggested by a study conducted on asymptomatic and symptomatic cases [6]. Interestingly, the kinetics of the humoral response was recently described in symptomatic (non-severe) and convalescent COVID-19 individuals and revealed that antibody levels against the receptor binding domain (RBD) of SARS-CoV-2 decline over time [7,8,9,10]. Despite these findings, long-term antibody responses remain to be fully characterized, particularly in hospitalized patients, and for instance, for those transferred to ICU and with very poor prognosis. In a context of serological surveillance, we sought to determine the kinetics of IgG, IgM, and IgA responses against SARS-CoV-2 in patients hospitalized at CHU of La Réunion (Centre Hospitalier Universitaire). Réunion Island, a French department, one of the outermost regions of Europe, has been impacted at the beginning of March 2020 by a first set of imported cases of SARS-CoV-2. On 29 August 2020, 1410 confirmed cases of COVID-19 have been reported on a total of about 850,000 inhabitants (WHO data).

We herein described serological results using a commercial and a robust conventional in-house cell-based ELISA for the detection of anti-SARS-CoV-2 Ig and assessed the evolution of Ig plasma levels over a two-month period in patients hospitalized in ICU versus non-ICU hospitalized cases.

## 2. Materials and Methods

### 2.1. Ethical Statement

This study was conducted at CHU of La Réunion (teaching hospital) which takes care of SARS-CoV-2-infected individuals originated from La Réunion, metropolitan France, and also from other territories of the Indian Ocean region, such as Mayotte.

Serological testing (collected as part of the patient’s care) was performed on plasma samples. Patients, at the time of entry at the hospital, consented for the testing (non-opposition and for research use only). Samples were deposited at the local Bioresource Centre (CRB, certified by Euro-Quality System, NF 996-900) and declared in the local bio-collection of infectious diseases. 

The date of disease onset was defined as the day when symptoms were perceived, based on patients’ declaration. For negative controls, plasma samples of healthy donors from the University and CHU of La Réunion were used under informed consent. The study was approved by the Human Ethics Committee of University of Bordeaux (“Comité Consultatif de Protection de Personnes se prêtant à des Recherches Biomédicales”, Bordeaux France, ref. 2008-A00151-54). The design of this work conforms to ethical standards of the Helsinki declaration. 

### 2.2. Study Design and Participants

Plasma samples from a case series of over 120 hospitalized patients with RT-PCR-confirmed COVID-19 were collected from March 2020. Among these samples, 91 were used to determine the positivity rate for anti-SARS-CoV-2 IgG according to the number of days after the onset of symptoms. In our cohort, twenty-six patients were followed-up for a longitudinal anti-SARS-CoV-2 Ig analysis taking into consideration their long hospital stay duration (at least two weeks) and the severity of the disease. Clinical outcomes were monitored until 26 June 2020, the final date of the follow-up. Patients who were admitted to the ICU (12 cases) presented severe complications, including acute respiratory distress syndrome (according to the Berlin definition [11]) or organ dysfunction. 

Data collection included demographic data, hospital stay duration, day of admission to ICU, oxygen therapy, comorbidities, and laboratory findings. For laboratory findings, each value corresponded to the maximum value of the measured parameter over the entire length of hospitalization. 

### 2.3. Commercial Enzyme-Linked Immunosorbent Assay (ELISA) to Test for Specific anti-SARS-CoV-2 IgG

The “Human COVID-19 IgG antibody Elisa kit” (MyBiosource, reference MBS3809906, San Diego, CA, USA) was used to detect anti-N (Nucleocapsid) IgG, according to the manufacturer’s instructions. Plasma samples were used at a 1:400 dilution.

### 2.4. In-House Cell-Based ELISA to Test for Specific anti-SARS-CoV-2 IgG, IgM, and IgA

Vero E6 cells were plated with Modified Eagle’s Medium (MEM eagle, PAN Biotech P0408500) supplemented with 10% heat-inactivated fetal bovine serum (FBS, PAN Biotech, 3302 P290907), 2 mM L-glutamine (Biochrom AG, K0282), 0.1 mg/mL penicillin–streptomycin (PAN Biotech, P0607100), 1 mM sodium pyruvate (PAN Biotech, P0443100), and 0.5 µg/mL amphotericin B (PAN Biotech, P0601001) in 96-well plates, at a density of 2 × 10^4^ cells/well. Cells were kept in biosafety level 3 facilities. At confluence, cells were either mock-infected or infected with a clinical isolate of SARS-CoV-2 (multiplicity of infection of 10). After three days of infection, plates were UV-irradiated (ten minutes), and washed with 0.9% NaCl solution before fixation in cold absolute ethanol (ten minutes). Plates were air-dried and stored at −20 °C until use. For anti-SARS-CoV-2 IgG detection, 96-well plates were first equilibrated for fifteen minutes at room temperature. Wells were rehydrated with 200 µL of 1X PBS for 30 min, then blocked with 200 µL of 5% non-fat dry milk-1X PBS for 1 h. Plasma samples were two-fold serially diluted (from 1:100 to 1:6400) with milk-1X PBS. Diluted samples (100 µL) were distributed in each well and incubated for one hour. Each well was then washed three times with 1X PBS containing 0.05% Tween-20 (PBS-T) and blocked with 200 µL of milk-PBS for 30 min. Mouse HRP-conjugated specific anti-human IgG (reference 9040-05), IgM (reference 9020-05), and IgA1/IgA2 (references 9130-05 and 9140-05) from Southern Biotech (Ozyme, France), at a 1:5000 to 1:10,000 dilution, were used as a secondary antibodies. Wells were washed four times with PBS-T, and 100 µL of 1X TMB substrate solution (eBioscience, Ozyme, France) were added. The reaction was stopped with 100 µL of 1M hydrochloric acid and the absorbance read at 450 nm with reference at 630 nm. Negative controls (plasma samples from seven healthy donors) were used to assess the background level. Samples having an absorbance greater than or equal to two times to that of the negative controls were considered as positive for anti-SARS-CoV-2 IgG. 

### 2.5. Western Blot (WB) Analyses to Determine the Nature of Viral Antigens Recognized by anti-SARS-CoV-2 IgG

For IgG Western blotting, non-infected and infected Vero E6 cells were lysed in RIPA lysis buffer (RIPA Pierce™ buffer, reference 89900, Thermo Fisher Scientific, Waltham, MA, USA). Wells of 10% SDS-PAGE Tris-Glycine homemade gel were charged with infected or non-infected Vero E6 cell lysates. Protein electrophoresis was performed at 30 mA per gel for 40 min. Proteins were transferred to a nitrocellulose membrane (Hybond-C Extra, Amersham Biosciences, reference RPN303E, UK) at 50 mA per gel for one hour and fifteen minutes. Equal cellular protein loads were confirmed by staining of the nitrocellulose membrane with a Red Ponceau solution (Appendix A). Membranes were next processed for WB analysis. After one hour of blocking with 5% non-fat dry milk−0.1% Tween 20-1X PBS, nitrocellulose membranes were incubated with 1:1000 dilution of patients’ plasma for one hour, washed three times with 0.1% Tween 20-1X PBS, then incubated with a 1:2,000 dilution of mouse anti-human IgG HRP-conjugated (SouthernBiotech, reference 9040-05). Finally, membranes were washed four times with 0.1% Tween 20-1X PBS. WB was revealed using either Vector^®^ NovaRED^TM^ Peroxidase Substrate (Vector Laboratories, reference SK-4800, USA) or enhanced chemiluminescence reagent (Amersham, reference RPN2232, UK).

### 2.6. Statistical Analyses

All statistical analyses were performed using GraphPad Prism 5 software. Categorical variables were presented as frequency rates and percentages, and continuous variables were described using either means and standard deviations (SDs) for normally distributed data, or medians and interquartile ranges (IQRs) for data that were not normally distributed. As recommended by GraphPad Statistics Guide, data that passed the D’Agostino–Pearson omnibus test and the Shapiro–Wilk test were considered as normally distributed and further tests were performed accordingly.

Proportions for categorical variables were compared using the Fisher exact test (two-tailed). Means for continuous variables were compared using independent group *t* tests (two-tailed) when the data were normally distributed; otherwise, the Mann–Whitney test (two-tailed) was used. Spearman rank correlation (one-tailed) was performed to measure the degree of association between the number of days after symptom onset and anti-SARS-CoV-2 IgG plasma levels. Statistical significance was set at the 0.05 probability level.

## 3. Results

### 3.1. Characteristics of the Cohort

#### 3.1.1. Determination of the Positivity Rate for anti-SARS-CoV-2 IgG according to the Number of Days after Symptoms

A cohort of 91 hospitalized COVID-19 patients (confirmed by RT-PCR) was followed at CHU of La Réunion, from the first reported cases in March 2020. Serologic features of this cohort are shown in Figure 1, presenting the graph of ELISA optical density (OD) values at 450 nm (reflecting anti-N IgG levels assessed by a commercial kit) versus days after symptom onset. A positive correlation was found between OD values at 450 nm and days after symptom onset (Spearman r = 0.52, *p*-value < 0.0001). OD values were under the cutoff detection for 39 patients on a total of 77 (51%) during the period ranging from day 1 to day 14. However, 100% of the patients tested from the fifteenth day after first symptoms were positive for anti-N IgG. 

#### 3.1.2. Demographic Data and Laboratory Findings of the Patients Selected for the Analysis of anti-SARS-CoV-2 IgG, IgM, and IgA Responses

In this cohort, 26 patients were admitted to isolation wards and selected for the analysis of anti-SARS-CoV-2 IgG, IgM, and IgA responses over time. They were enrolled in the study given that they were all hospitalized for at least two weeks. The mean of age of the study population was 64 years (SD, 16), and 81% were men. Of these patients, 12 (46%) were transferred to the ICU because of respiratory complications (Appendix A). Laboratory findings of ICU and non-ICU patients included hematological and biochemical data (Appendix A). Of note, the other 65 patients recovered from the disease and were discharged from the hospital before the fifteenth day after the first symptoms.

### 3.2. Kinetics of anti-SARS-CoV-2 Antibody Responses in non-ICU and ICU Patients

#### 3.2.1. Kinetics of anti-SARS-CoV-2 IgG Response in non-ICU and ICU Patients

We first investigated IgG plasma levels in ICU patients using an in-house cell-based ELISA. For each of these patients, at least two plasma samples collected on two separate days were obtained during the period starting from day 15 after the first symptoms. Patients 1 to 9 had no medical history of hematological malignancies. Their IgG kinetics accompanied by a linear regression curve were depicted on the same graph (Figure 2A). Almost all the slopes calculated from linear regression curves (8/9, 89%)) were negative, indicating a declining anti-SARS-CoV-2 IgG response in the severe ICU forms of COVID-19. Of further note, patient 10 was affected by a chronic lymphocytic leukemia and patient 11 by Waldenström macroglobulinemia. Anti-SARS-CoV-2 IgG were detectable in patient 10, but at very low levels (OD values < 0.7 up to 44 days after symptom onset) (Figure 2A). In patient 11, IgG levels remained below the cutoff detection rate up to 23 days after symptom onset. This is in contrast with the positive correlation previously established in Figure 1, showing a 100% positivity rate for anti-N IgG from the fifteenth day after symptom onset. These two patients, therefore, and as expected, have a very poor global humoral immunity against SARS-CoV-2.

In regards to IgG kinetics in non-ICU patients (seven cases analyzed), six slopes (6/7, 86%) calculated from linear regression curves were positive (Figure 2B). As featured in Figure 2C, a comparative analysis performed on the slopes of IgG kinetics between both groups showed a significant lower value in the ICU versus the non-ICU group (−0.07, IQR (−0.1–(−0.02)) versus + 0.04, IQR (0.003–0.07)). This result further emphasizes that plasma IgG levels against the virus are dramatically reduced over time in severe ICU forms of the disease.

Two patients (one ICU and one non-ICU) were further analyzed using our in-house cell-based ELISA and compared to the commercial ELISA for nucleocapsid. For each patient, five samples were collected over of a period of up to 60 days. 

Our results using the commercial ELISA showed that, in the non-ICU patient, anti-N IgG levels continuously increased over time, i.e., between day 19 and day 57 after the first symptoms (Appendix A). At the 1:400 dilution, the OD value at 450 nm was equal to 0.16 at day 19, and to 0.71 at day 57 (Appendix A). In the ICU patient, plasma anti-N IgG levels reached a peak during a first phase (between day 7 and day 12 after symptom onset), and then markedly decreased during a second phase (between day 12 and day 40 after symptom onset). Indeed, the OD value was equal to 0.69 at day 12, and to 0.23 at day 40 (Appendix A). 

Our in-house cell-based ELISA revealed that, in the ICU patient and at the 1:400 dilution, the OD value at 450 nm was equal to 2.4 at day 12, and to 1.2 at day 40 (Appendix A). At the same dilution, in the non-ICU patient, the OD value was equal to 0.9 at day 19, and to 2.5 at day 57 (Appendix A). Of note, with the commercial kit, all OD values were lower than those obtained with our in-house cell-based ELISA (OD values < 1). This is to be expected given that our method allows for the detection of plasma IgG directed against various viral protein epitopes exposed by Vero E6 cells, and not solely on epitopes of the nucleocapsid protein.

#### 3.2.2. Determination of the Viral Antigens Recognized by anti-SARS-CoV-2 IgG

To determine the nature of the viral antigens recognized by anti-SARS-CoV-2 IgG, a Western blot analysis of either mock- or SARS-CoV-2-infected Vero E6 cells was performed on plasma samples of the ICU patient and the non-ICU patient. Whatever the plasma dilution used (1:100 or 1:1000), IgG reactivity was against a major protein with a molecular weight around 50 kDa in both patients (Appendix A). This likely corresponded to IgG recognizing the SARS-CoV-2 nucleocapsid protein [12]. At a 1:100 dilution, a detectable reactivity was also observed against the Spike protein of SARS-CoV-2 (bands migrating at around 125 kDa) and its S1 (bands between 93 and 125 kDa) and S2 (bands at about 93 kDa) subunits (Appendix A). While the IgG response against the Spike protein remained weak and with no major fluctuation in the reactivity over time in the ICU patient (Appendix A), this response clearly increased in the non-ICU patient from day 19 to day 57 after symptom onset (Appendix A). At a 1:1000 dilution, the variation of the IgG response against the nucleocapsid was much more apparent and confirmed the kinetics previously observed using the commercial ELISA (anti-N IgG), i.e., a decline over time in the ICU patient (Appendix A) and an increase in the non-ICU patient (Appendix A). Of note, similar amounts of total protein loaded per well were checked by staining nitrocellulose membranes with a solution of Ponceau Red (Appendix A).

#### 3.2.3. Kinetics of anti-SARS-CoV-2 IgM and IgA Responses in non-ICU and ICU Patients

By using our in-house cell-based ELISA, we also assessed specific IgM and IgA plasma levels in ICU and non-ICU patients. No major differences were evidenced for IgM kinetics between both groups (Figure 3A,B). In contrast to non-ICU patients, low levels of plasma IgA were measured in all ICU patients over time with OD values close to the cutoff (Figure 3C,D).

## 4. Discussion

During the early phase of the COVID-19 pandemic, confirmed cases were mostly imported among travelers from Wuhan (China) [13]. In France, three cases of COVID-19 were confirmed on 24 January 2020 as the first cases in Europe [14]. On March 2020, the first COVID-19 cases were imported from metropolitan France to Réunion Island. In tropical areas where arboviruses and COVID-19 coexist, clinical diagnosis and differential serological surveillance are critically important. Recently, a case of co-infection of dengue and COVID-19 was reported in Réunion Island, highlighting the importance of population surveillance by individual serological assays [15]. Serological surveillance is also an essential resource to evaluate the antibody response against specific antigens, and hence, to define vaccine programs.

In the present work, we aimed to analyze long-lasting IgG responses against SARS-CoV-2 in patients hospitalized in ICU (versus non-ICU), over time. We also analyzed the kinetics of IgM and IgA antibody responses which are both known to precede the IgG response, to be short-lived but yet essential to control the early phase of the infection.

We herein also provided the technical flow of a robust ELISA technique for the detection of anti-SARS-CoV-2 IgG, based on infected Vero E6 cells fixed and permeabilized in 96-well plates. Our results are in accordance with a commercial ELISA kit relying on the detection of IgG against the nucleocapsid. Interestingly, both assays revealed a very contrasted IgG response between non-ICU hospitalized patients and ICU patients, over a two-month period. While a continuous increase in specific IgG levels was observed in non-ICU patients, ICU patients presented a peak (within the two first weeks after symptom onset) followed by a marked decrease (from the third week after symptom onset). Anti-N IgG appeared to contribute to the kinetics. Our data are complementary to those of a previous study showing different profiles of IgG and IgM kinetics (against the viral nucleocapsid and the receptor binding domain of S1) over a maximum one-month period after symptom onset. Indeed, a continuous increase in antibody serum levels was found in mild cases, whereas this pattern began to be altered (no increase) in severe (ICU) cases [16]. Other investigations also corroborate our results. Two studies conducted on SARS-CoV-2 positive individuals in England revealed a significant decline of neutralizing antibody responses in the three months following infection [17,18]. In another one, Long et al. argued that a proportion of symptomatic COVID-19 patients (12.9%) became seronegative for IgG in the early convalescent phase (i.e., eight weeks after they were discharged from the hospital) [6]. Such findings emphasize the need to have major follow-up studies and, for instance, to collect and store acute and convalescent serum samples to characterize the immune response against SARS-CoV-2, as recommended by current guidelines from the WHO. Although other studies from other countries will be useful to confirm these results, the current data argue that the immune protection against SARS-CoV-2 may rapidly vanish over time (in a matter of a few weeks) in COVID-19 cases, particularly in severe cases. 

An altered antibody response during the course of COVID-19 could be one explanation to the fact that after discharge from hospital, some patients remain/return viral RT-PCR positive and others may even relapse [19,20]. This was not described for severe SARS-CoV-1 patients who had a robust IgG response within the first four weeks of illness [21]. However, among patients who recovered from SARS-CoV-1, a marked decrease of IgG levels was measured over time, but, importantly, only after sixteen months following disease onset [22]. Critically, in our study regarding SARS-CoV-2 patients, the decline was much faster and appeared in a matter of weeks and not months. The mechanisms involved in the rapid decrease in IgG levels observed in our ICU patients during COVID-19 remain to be elucidated. One explanation for this outcome could be a reduced B cell count in severe cases, as highlighted by several studies, including a meta-analysis of lymphocyte subset counts in COVID-19 patients [23,24,25]. A suboptimal T cell response may be also involved in this outcome. Indeed, it was demonstrated that binding and neutralizing antibody responses decay over the first four months post-infection with a similar decline in Spike-specific CD4+ and circulating T follicular helper frequencies in a cohort of patients who recovered from mild/moderate COVID-19 [26]. Additional analyses are required, particularly addressing B cell activation status to further characterize the humoral response according to COVID-19 severity. As suggested by Shi and collaborators, it would be of interest to determine whether specific major-histocompatibility-complex antigen loci (HLA) are associated with the development of unique and unusual anti-SARS-CoV-2 immunity [19].

Another novel and important finding is that ICU patients failed to have a robust specific IgA response against the virus. IgA is the major antibody to engage defense functions at different anatomic levels in relation to the mucosal epithelium. IgA is thought to be able to interact with intracellular pathogens such as viruses, blocking replication, assembly, and/or budding. Our findings would argue that ICU patients may be impaired in their capacity to control SARS-CoV-2 infection, for instance, at the level of the lung and nasopharynx epithelia.

Our original data have several major implications. Our results clearly argue for the use of conventional methods to perform reliable serological surveillance and particularly given the major shortage of commercial kits. Our assay is robust, given that the detection is against several viral antigens and the use of different monoclonal anti-human Ig antibodies allowed us to screen for IgG, IgM, and IgA responses. Serological assays may be helpful for the diagnosis of suspected patients with negative RT-PCR results and for the identification of asymptomatic infections. Medical care and protections should be maintained, particularly for recovered patients (severe cases) who may remain at risk of reinfection. 

This study has some limitations. The neutralizing activity of plasma samples is a useful marker associated with COVID-19 severity, particularly at earlier infection stages [27], and this should be determined. This could be ascertained by measuring the IgG response specifically against the receptor binding domain of SARS-CoV-2 Spike protein. Experiments along these lines are ongoing in our laboratory.

In conclusion, this report emphasizes the need to develop strong vaccine approaches, as immunity that occurs naturally during COVID-19 is not optimal in all individuals (short-lived in severe cases). Passive immunization that does not require the activation of immune cells may provide a direct response against the virus [28,29]. However, this would be an emergency strategy and not a substitute for highly warranted vaccine strategies. In any case, preventive measures, such as social distancing, universal public masking, and application of standards of hygiene should be maintained to limit SARS-CoV-2 propagation [30,31].

## Figures and Tables

**Figure 1 jcm-09-03847-f001:**
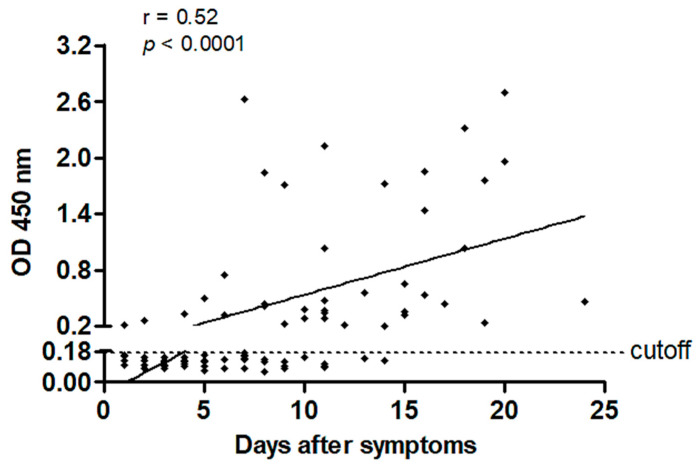
Association between anti-N (nucleocapsid) IgG plasma levels (optical density (OD) 450 nm) and the number of days after symptom onset in coronavirus disease 2019 (COVID-19) patients hospitalized at CHU of La Réunion. A positive correlation was evidenced between these two parameters. Linear regression (solid line) is depicted in the plot. Spearman correlation coefficient (r), and *p*-value (one-tailed) are depicted.

**Figure 2 jcm-09-03847-f002:**
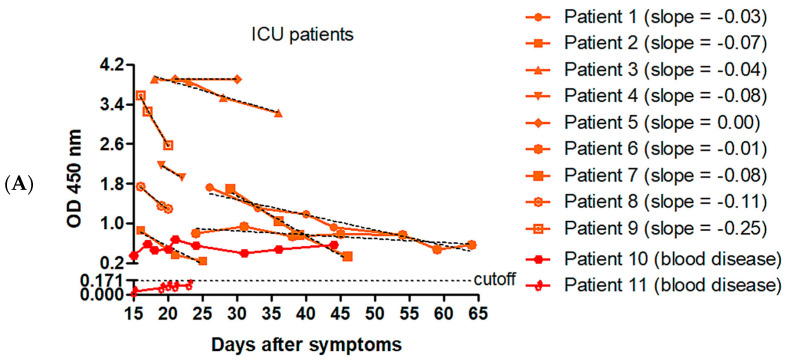
Kinetics of IgG response to SARS-CoV-2 in intensive care unit (ICU) (*n* = 11) and non-ICU patients (*n* = 7) hospitalized for at least two weeks. (**A**) By using the in-house cell-based ELISA, the kinetics of IgG response was studied in ICU cases, from day 15 after symptom onset. Patients 1 to 9 had no medical history of hematological malignancies. Patient 10 was affected by a chronic lymphocytic leukemia and patient 11 by Waldenström macroglobulinemia. Regression linear curves (dotted black lines) and calculated slopes are represented. (**B**) A similar analysis was performed in non-ICU cases. (**C**) Slopes of linear regression curves for both groups were compared. Median, interquartile range (IQR), and *p*-value are represented.

**Figure 3 jcm-09-03847-f003:**
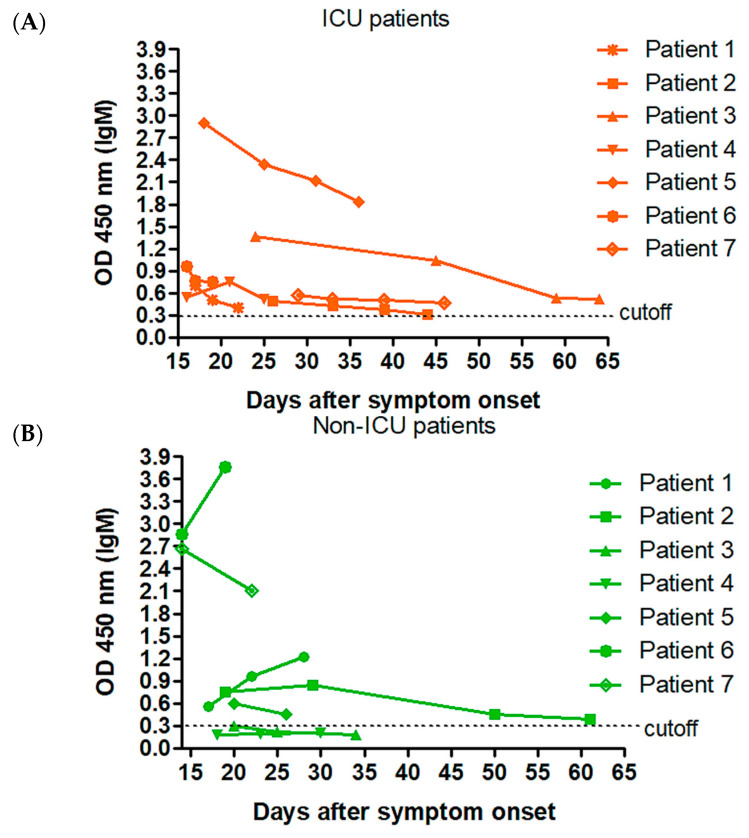
Kinetics of IgM (**A**,**B**) and IgA (**C**,**D**) responses to SARS-CoV-2 in ICU (*n* = 7) and non-ICU patients (*n* = 7) with no medical history of hematological malignancies and hospitalized for at least two weeks. By using the in-house cell-based ELISA, the antibody response was studied in ICU cases (**A**–**C**), from day 15 after symptom onset. A similar analysis was performed in non-ICU cases (**B**–**D**).

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
