# Peer review of "Serological Surveillance of COVID-19 Hospitalized Patients in Réunion Island (France) Revealed that Specific Immunoglobulin G Are Rapidly Vanishing in Severe Cases"

_jcm, 2020, doi:10.3390/jcm9123847_

Round 1

Reviewer 1 Report

This is an interesting manuscript reporting on IgG responses against the N protein (commercial assay) and SARS-CoV-2 infected cells using an elegant cell-based ELISA.  The authors analyzed a small cohort of ICU and non-ICU patients and observed weak IgG levels (as specified above) for ICU patients.  This is potentially an important finding; however several technical aspects of the work need to be addressed.

Comments

1- The humoral responses are larger than IgG.  The authors must also report on, at least, IgA and IgM.  This could be easily addressed by changing the secondary antibody used in the ELISA assays.

2- Several reports have shown that all isotypes (IgG, IgA, IgM) contribute to SARS-CoV-2 neutralization by plasma from recovered individuals.  Thus highlighting the need to measure these responses (point #1, above).

3- What about neutralization?  The authors propose at lines 313-314 to “measuring the IgG responses specifically against the receptor binding domain of SARS-CoV-2 Spike protein”, however, this is only a surrogate and viral neutralization must be reported.  If not possible, then at least ELISA against the RBD domain for IgG, IgA and IgM must be shown. In other words, saying that no neutralization was measured in the current study is a limitation, it is NOT sufficient.  Some data should be shown.

4- There appears to be a confusion between “plasma” and “sera”.  For example, Figure 1 describes “plasma” levels, Figure 2A “dilution of sera”, Figure 2C “dilution of plasma”.  However, in the Material and methods section only “plasma” is indicated (line 68).

5- it is unclear how the 1:400 (for commercial ELISA), 1:1000 (cell-based ELISA) and 1:100 (WB) dilutions were selected.

6- A loading control is missing in Figure 4.  It is indicated that 10ul of infected or non-infected cells were loaded on a gel…but this doesn’t mean anything unless we have a control showing that similar amounts of total proteins were loaded.

7-  Several statements are inaccurate:

Abstract: “Little is known about the kinetics of the antibody response against severe acute respiratory syndrome coronavirus 2 (SARS-23 CoV-2)…” we have gained a formidable knowledge about the kinetics of antibody responses in the last few months and the authors need to do a better work to acknowledge this.

At the bare minimum the following manuscripts should be referenced (certainly at lines 49-53) and discussed:

https://www.medrxiv.org/content/10.1101/2020.07.09.20148429v1

https://pubmed.ncbi.nlm.nih.gov/33015650/

https://pubmed.ncbi.nlm.nih.gov/33001206/

https://www.biorxiv.org/content/10.1101/2020.07.09.194639v1

https://www.medrxiv.org/content/10.1101/2020.09.09.20191205v2

https://www.mdpi.com/2077-0383/9/10/3188

8- line 44, please specify and provide updated references… certainly not 15% of the 31M of infected individuals progressed to a severe form of the disease.  Additional references with more accurate % should be provided.

9- What the authors call an “in house ELISA” is an “in house cell-based ELISA”

Reviewer 2 Report

Authors described serological results of 91 COVID-19 patients using a commercial and a robust conventional in-house ELISA for the detection of anti-SARS-CoV-2 IgG and assessed the evolution of IgG plasma levels over a two-month period in patients hospitalized in ICU versus non-ICU hospitalized cases. For negative controls, plasma samples of healthy donors were sampled. Results showed that among the 91 included patients, 100% of them showed positive IgG after 15 days from symptoms onset, while 51% from day 1 to 14 and the ELISA test was correlated with the days after symptom onset.

Major issues:

  • The manuscript is well written, and the methods appropriated; however, clarity s needed concerning the main outcome.
  • You initially describe clinical and demographic characteristics of 91 patients, focusing on IgG findings, you therefore select 26 patients (which is the reason? Please clarify), and you finally focus the entire paper, answering to the main question you raised, including two only patients.
  • It is not clear the real population: it seems that among 91 patients with COVID-19, 12 are admitted to ICU and 14 are admitted to another ward. What about the other 65? Please clarify in the text. Moreover, although demographic data are essential, these informations should be limited to a table and not repeated in the text (the main outcome of the study is not to describe patients’ characteristics), since it may deflect attention from the primary outcome. Otherwise the results seem confused. The same for laboratory findings that should be moved to a Supplemental File.
  • The manuscript should be right from the beginning. You should focus on the two patients you extensively described for IgG findings. I suggest focusing on the other cohorts in a second phase, maybe on a Supplemental File. Otherwise the paper is very confused.
  • Discussion: I would reorganize following the same structure as above.

Minor comments:

  • The absorbance value (OD) of ELISA is not defined in the text
  • Figure 1, I suggest amplifying the y axis from 0 to 0.2 to reduce the overlaps
  • “For negative controls, plasma samples of healthy donors were sampled”. I do not find these data.

Round 2

Reviewer 1 Report

The revised paper has been greatly improved.  I understand the time and efforts required to perform the neutralization experiments and therefore I do not request any longer to be performed.

Some of the suggested references in my first review appear to have been published in good peer-reviewed journals and I strongly encourage the authors to include them as they are directly on the topic of their manuscript.

Author Response

Dear Reviewer,

Thank you for your comments. All suggested references were added in the manuscript (line 70, line 553 and line 575).

Reviewer 2 Report

Thank you for providing a new version incorporating most of the suggestions I made. I would only to clarify in the methods the first and second aims, and to use Titles and subtitles in the results to answer each aim and make it more clear to the reader. Some minor English and "format" errors need to be checked. 
